Cryptic diversity on the genus Caenolestes (Caenolestidae: Paucituberculata) in the Ecuadorian Andes

Carrión-Olmedo Julio C. julio.carrion@biodiversidad.gob.ec
Brito Jorge
Instituto Nacional de Biodiversidad , Quito , Ecuador
Manjarrez Javier
Electronic publication date: 2025 Jul 10
Publication date: 2025
Volume: 13
Electronic Location ID: e19648
Received 2024 Nov 22; Accepted 2025 Jun 2
Copyright: ©2025 Carrión-Olmedo and Brito
Copyright year: 2025
Copyright holder: Carrión-Olmedo and Brito
License: This is an open access article distributed under the terms of the Creative Commons Attribution License, which permits unrestricted use, distribution, reproduction and adaptation in any medium and for any purpose provided that it is properly attributed. For attribution, the original author(s), title, publication source (PeerJ) and either DOI or URL of the article must be cited.
License URL: https://creativecommons.org/licenses/by/4.0/

Keywords: Marsupials, Species complex, Andean biodiversity, Molecular phylogenetics

Funding: Gobierno Autónomo Descentralizado El Oro MacArthur Foundation Earth Deeds Carbon Mitigation Initiative of the Pacific Lutheran University Fundación Aves y Conservación Fundación Ecominga This work was supported by Gobierno Autónomo Descentralizado El Oro, the MacArthur Foundation, Earth Deeds Carbon Mitigation Initiative of the Pacific Lutheran University, Fundación Aves y Conservación, and Fundación Ecominga. The funders had no role in study design, data collection and analysis, decision to publish, or preparation of the manuscript.

==============================
The shrew-opposums of the genus Caenolestes (Thomas, 1895) belong to a group of Ameridelphian marsupial mammals in the order Paucituberculata, an order in which most of its representatives are extinct. This genus contains five formally described species: C. caniventer, C. convelatus, C. condorensis, C. fuliginosus, and C. sangay, all with type localities in Ecuador, plus at least three candidate species from Colombia now recognized as subspecies C. convelatus barbarensis, C. fuliginosus centralis and C. fuliginosus obscurus. Records of this genus are not abundant, both in biological collections and in sequence repositories (GenBank); thus, showing a discontinuous geographical distribution that could be a consequence of incomplete sampling. Systematic expeditions by the Instituto Nacional de Biodiversidad (INABIO) have increased the geographic sampling of this genus in Ecuador, which allowed us to reevaluate its genetic diversity. We obtained 43 sequences of cytb and 30 of RAG1 from 28 Ecuadorian localities in 12 Provinces, including novel topotypical material from C. caniventer, C. convelatus, and C. condorensis. We present a new hypothesis on the genetic diversity of Caenolestes using maximum likelihood inference for phylogenetic analysis, estimate p-genetic distances and divergence times for the genus. We found a species complex in the C. fuliginosus clade, with at least three candidate new species, having a threshold above 5% for the estimated genetic distance of the cytb among them. Also, we found two additional lineages hidden within C. caniventer. We expect that future work, with similar or larger sampling efforts in Colombia and Peru would reveal greater phylogenetic diversity and more complete evolutionary relationships.

Introduction

Paucituberculata is a group of shrew-like basal metatherians that radiated in South America (Abello, Toledo & Ortiz-Jaureguizar, 2018) and is considered a relictual element in modern faunas (Voss & Jansa, 2009). This group’s evolutionary history has been drastically shaped throughout the Cenozoic, from being a diverse group in the Eocene and Oligocene to experiencing significant extinctions in the late Miocene that have led to only a few extant species (Pardiñas et al., 2017; Abello, Martin & Cardoso, 2021; Voss & Jansa, 2021). While the evolutionary past of Paucituberculatans is well documented (e.g., Abello, Toledo & Ortiz-Jaureguizar, 2018; Abello, Martin & Cardoso, 2021), the last systematic revision of the extant species was over a decade ago by Ojala-Barbour et al. (2013), and evolutionary history of extant species has never been assessed in detail.

Paucituberculata is a monotypic order with Caenolestidae (Trouessart, 1898) as the only representative family and three genera: Caenolestes (Thomas, 1895), Lestoros (Oehser, 1934), and Rhyncholestes (Osgood, 1924). Caenolestes is the most diverse, with five recognized species (Tomes, 1863; Anthony, 1921; Anthony, 1923; Anthony, 1924; Albuja & Patterson, 1996; Ojala-Barbour et al., 2013), and at least three candidates for Colombia (González-Chávez, 2024). Few works on Caenolestes have been published to understand its natural history (e.g., Brito & Ojala-Barbour, 2016; Martin & González-Chávez, 2016; Brito et al., 2023) as well as other morphological (e.g., Osgood & Herrick , 1921; González-Chávez, Rojas-Díaz & Cruz-Bernate, 2019; Brito et al., 2022a; Brito et al., 2022b; González et al., 2024) and developmental (e.g., González et al., 2020) aspects.

All caenolestids are distributed in cold and humid environments in the Andes from Venezuela to Perú (Patterson, 2008). Anthony (1924) recognized two morphologically and ecologically distinct groups within Caenolestes. Subsequently, Albuja & Patterson (1996), Timm & Patterson (2008), Ojala-Barbour et al. (2013) and González et al. (2024) adopted this proposal, but without testing it. The first group consisted only of C. fuliginosus, a small, gracile, and delicate shrew-opossum with a wide distribution from paramo to high elevation cloud forests, and a second group represented by C. convelatus, C. condorensis, C caniventer, and C. sangay that are large shrew-opossum restricted to middle elevation cloud forest and subtropical habitats. From a phylogenetic approach, Ojala-Barbour et al. (2013) noted that these ecological groups are not recovered as monophyletic, as C. convelatus is the most basal species of the genus. They recognized limitations of their research due to sampling size and suggested that systematic sampling efforts and additional genetic data may help to understand Caenolestes. Their hypothesis lacked topotypical sequences of C. caniventer and C. condorensis; therefore, an updated revision was needed to shed light on this genus. This group have been particularly difficult to assess phylogenetically as it exemplifies numerous challenges to fully understand its diversity: (1) rarity in field surveys, (2) little availability of museum specimens difficult to extract DNA from, (3) may exhibit cryptic diversity, and (4) conservative morphological characters, thus, extensive surveys and collection of genetic data are much needed.

Recently, González et al. (2024) presented an extensive morphological and ecological review of Colombian paucituberculatans and showed evidence to recognize the subspecies suggested by Bublitz (1987): Caenolestes convelatus barbarensis, C. fuliginosus centralis, and C. f. obscurus as species. However, it is still necessary to phylogenetically assess these findings. On the other hand, in Ecuador, the Instituto Nacional de Biodiversidad made additional sampling efforts during the last decade and accumulated a comprehensive collection of caenolestids throughout Ecuador. Here, we include novel topotypical sequences of Caenolestes caniventer, C. convelatus, and C. condorensis, present an extensive phylogenetic revision of Ecuadorian taxa, discover unexpected diversity hidden within populations of C. fuliginosus, and estimate divergence times for the group.

Materials & Methods

Studied specimens

Livetrapping was conducted throughout the Andes during a systematic sampling effort by the Instituto Nacional de Biodiversidad (INABIO) that includes the last decade. Most of the Ecuadorian specimens studied were collected by JB author and collaborators during field trips conducted in the Reserva Dracula, Parque Nacional Sangay, Cordillera de Chilla, Reserva Geobotánica Pululahua, Corredor Ecológico Llanganates-Sangay, Parque Nacional Cotacachi-Cayapas, Bosque Protector Golondrinas, Tandayapa Cloudforest Station USFQ and Reserva Privada Sabia Esperanza. These surveys involved a cumulative trap effort of 15,000 trap/nights. Capture and handling, euthanasia, and subsequent preservation of field secured specimens followed the guidelines established by the American Society of Mammalogists (Sikes & Animal Care Use Committee of the American Society of Mammalogists, 2016). The collection permits issued by Ministerio del Ambiente, Agua y Transición Ecológica del Ecuador (MAATE) that allowed the study to be carried out are as follows: No. 005-2014-I-B-DPMS/MAE, 007-IC-DPACH-MAE-2016, 005-IC-FLOFAU-DPAEO-MAE, 003-2019-IC-FLO-FAU-DPAC/MAE, MAE-DNB-CM-2019-0126, MAAE-ARSFC-2020-0642, MAAE-ARSFC-2021-1644MAATE-ARSFC-2023-0145, MAATE-ARSFC-2024-1064, and the authorization for access to genetic resources No. MAATE-DBI-CM-2023-0334. Specimens were deposited in the mammalogy collection at the Instituto Nacional de Biodiversidad (MECN), Quito, Ecuador. Additionally, the Museo de Zoología de la Universidad del Azuay (MZUA), Cuenca, Ecuador, and the Museo de Zoología QCAZ from the Pontificia Universidad Católica del Ecuador PUCE, Quito, Ecuador provided invaluable specimens to complete geographic sampling.

Tissue sampling

Liver and muscle tissues were preserved in 96% ethanol. We selected at least one adult specimen of the collected series at each locality. We procured sample type localities of each Caenolestes species. Material used for molecular analysis is described in Table 1.

Table 1 Species, vouchers, and GenBank accession numbers for newly generated DNA sequences used in genetic analyses.

Species	Voucher ID	Cytb	RAG1	Citation	
Rhyncholestes raphanurus	n/a	AJ508399.1		Nilsson et al. (2003)	
Lestoros inca	FMNH 174481	KF418779.1		Ojala-Barbour et al. (2013)	
	n/a	U34681		Patton, dos Reis & da Silva (1996)	
Caenolestes convelatus	QCAZ 1848	KF418782.1		Ojala-Barbour et al. (2013)	
	QCAZM 16704	PV505064	PV505079	This paper	
	QCAZM 13017	PV505063	PV505078		
Caenolestes condorensis	MECN 3726	PQ570945		This paper	
	QCAZM 18923	PV505065	PV505080		
Caenolestes sangay	MEPN 12137	KF418781.1		Ojala-Barbour et al. (2013)	
	MECN 4330	PQ570946	PV505082	This paper	
	MECN 4349	PQ570947	PV505081		
	MECN 4331	PV505066	PV505083		
	MECN 5665		PV505084		
	MECN 4346		PV505085		
Caenolestes caniventer	MECN 4811	PQ570948	PV505091	This paper	
	MECN 4812	PQ570949			
	MECN 4900	PQ570950	PV505089		
	MECN 6938	PQ570951	PV505090		
	MECN 7018	PQ570952			
	MECN 7729	PQ570953	PV505088		
	MECN 7918	PQ570954			
	MZUA 0338	PQ570955			
	QCAZM 17851	PV505067			
Caenolestes fuliginosus	n/a	AJ508400.1		Nilsson et al. (2003)	
	MECN 5231	PQ570956		This paper	
	MECN 5242	PQ570957	PV505105		
	MECN 5268	PQ570958			
	MECN 5269	PQ570959	PV505104		
	MECN 5273	PQ570960	PV505103		
	MECN 6058	PQ570961	PV505102		
	MECN 6059	PQ570962			
	MZUA 0344	PQ570963			
	MECN 8302	PV505068			
	MECN 8303	PV505069	PV505106		
	MECN 8304	PV505070	PV505107		
	MECN 4940	PV505071			
	QCAZM 11304	PV505072			
Caenolestes sp. 1 Loja	QCAZM 16357	PV505073	PV505086	This paper	
	QCAZM 15516	PV505074	PV505087		
Caenolestes sp. 2 Perú	AMNH 268103	KF418780.1		Ojala-Barbour et al. (2013)	
Caenolestes sp. 3 Baños	MECN 6424	PQ570964	PV505093	This paper	
	MECN 7239	PQ570965			
	MECN 7682	PQ570966	PV505092		
Caenolestes sp. 4 Carchi	MECN 7838	PQ570967	PV505094	This paper	
	MECN 8042	PV505075	PV505095		
Caenolestes sp. 5 Imbabura	MECN 7811	PQ570968	PV505101	This paper	
	MECN 7812	PQ570969	PV505100		
	MECN 7816	PQ570970	PV505099		
	MECN 7880	PQ570971	PV505096		
	QCAZM 8400	PV505076	PV505097		
	QCAZM 8374	PV505077	PV505098		
Caenolestes sp. 6 Toisán	MECN 3745	PQ570972		This paper	
Notes.

Acronyms MECN División de Mastozoología, Instituto Nacional de Biodiversidad, Quito

QCAZM Museo de Zoología, Pontificia Universidad Católica del Ecuador

MEPN Escuela Politécnica Nacional, Quito

MZUA Museo de Zoología, Universidad del Azuay, Cuenca

FMNH Field Museum of Natural History, Chicaco

AMNH American Museum of Natural History, New York

Sequencing and bioinformatics

gDNA extraction, PCR amplification and Nanopore sequencing were made at the Laboratory of Nucleic Acids of the Instituto Nacional de Biodiversidad (INABIO) in Quito, Ecuador. DNA was extracted from bland tissue under the manufacturer’s protocol using the GeneJET Genomic DNA Purification Kit (K0722).

Two fragments were amplified by Polymerase Chain Reaction (PCR). The mitochondrial Cytochrome b (cytb) and the nuclear Recombination Activating Gene 1 (RAG1). 10 µl PCR reactions consisted of 4.9 µl of 2X Dreamtaq Hotstart Mastermix (ThermoFisher), one µl of DNA template (10 ng), 0.8 µl of Forward primer (10 µM), 0.8 µl of Reverse primer (10 µM), and 2.5 µl of nuclease-free water. We used universal primers: MTCB-F (5′-CCHCCATAAATAGGNGAAGG-3′) and MTCB-Rc (5′-WAGAAYTTCAGCTTTGGG-3′) to amplify cytb (Naidu et al., 2012) and RAG-didFor1 (5′-GTACCAGATGAAATCCAGTACCCA-3′) and RAG-didRevO (5′-TACCAGATTCATTCCCTTCACT -3′) for RAG1 (Gruber, Voss & Jansa, 2007). PCRs were performed on a miniPCR thermocycler under the following cycle profiles. For Cytochrome b, initial denaturation at 95 °C for 180 s, denaturation at 95 °C for 120 s, annealing at 45 °C for 30 s, extension at 72 °C for 80 s, with 32–35 cycles and a final extension at 72 °C for 10 min. For RAG1, initial denaturation at 95 °C for 180 s, denaturation at 95 °C for 45 s, annealing at 57 °C for 45 s, extension at 72 °C for 4 min, with 35 cycles and a final extension at 72 °C for 10 min.

Amplicons were sequenced in a minION Mk1C with a Flongle Flow Cell R10.4.1, and the Rapid Barcoding kit 96 V14 (SQK-RBK114.96). Data was basecalled and demultiplexed under Super-accurate basecalling v4.3.0, 400 bps model with the Dorado 7.6.7 package. Consensus sequences were generated with NGSpeciesID (Sahlin, Lim & Prost, 2021) using a Q12 threshold for FASTQ reads.

We chose super-accurate basecalling as this model produces modal raw read accuracies of 98.3% in comparison to other models of basecalling as High-accuracy (97.8% raw read accuracy) or Fast basecalling (96% raw read accuracy). Additionally, all consensus sequences produced had between 100–1,000 supporting reads, with an expected divergence from Sanger between 0.00–0.04% (Vasiljevic et al., 2021). This level of divergence is considered negligible for phylogenetic implications.

FASTQ output management, consensus generation, and subsequent FASTA data management, renaming, and concatenation were automated using a custom python script available in Zenodo by Carrión-Olmedo (2024).

Phylogenetic analyses

We performed two main phylogenetic inference analyses. The first analysis consisted of a codon-partitioned concatenated matrix of cytb and RAG1. The second analysis consisted of a concatenated matrix of the amino acids of cytb and RAG1. Also, we run four additional analyses as follows: codon partitioned DNA matrix of cytb, codon partitioned DNA matrix of RAG1, translated matrix of cytb, and translated matrix of RAG1.

The matrices were built with the newly generated sequences and those available in GenBank for the cytb and RAG1 of all paucituberculatans, originally released by Patton, dos Reis & da Silva (1996), Nilsson et al. (2003), Meredith et al. (2011), and Ojala-Barbour et al. (2013). The DNA character matrices were interpreted as coding sequences CDS and color-coded by amino acids in Mesquite 3.81 (Maddison & Maddison, 2023), aligned using the MAFFT algorithm 7.526 (Katoh et al., 2002; Katoh & Standley, 2013), and visually inspected for unambiguous errors. Then the DNA matrices were translated to amino acids for downstream analysis. The matrices and the partitions are available as dataset in Zenodo by Carrión-Olmedo & Brito (2025).

We chose maximum likelihood (ML) as the optimality criterion and used W-IQ-TREE 2.3.4 (Trifinopoulos et al., 2016) to determine the best-fitting evolutionary model for partition, perform phylogenetic inference, and evaluate branch support. The evolutionary model for each partition was evaluated using the -m test (Chernomor, von Haeseler & Minh, 2016). The nucleotide and amino acids ML trees were inferred using default parameters. Branch support was estimated with ultrafast bootstrap from 5,000 pseudorreplicates and 1,000 iterations as a maximum threshold to stop the search (-bb 5,000), and SH-like approximate likelihood ratio test (SH-aLRT) with 1,000 replicates (-alrt 1,000), support values mentioned herein follows this format SH-aLRT support (%)/ultrafast bootstrap support (%) (Hoang et al., 2018).

To estimate the uncorrected pairwise genetic p-distances, we used the cytb character matrix and obtained a pairwise table of distances with the Molecular Evolutionary Genetics Analysis (MEGA), version 11 (Tamura, Stecher & Kumar, 2021). The obtained distance matrix was plotted as a heat map in R 4.2.1 (R Core Team, 2022), with the ‘ggplot2’ (Wickham et al., 2023a; Wickham et al., 2023b) and ‘dplyr’ (Wickham et al., 2023a; Wickham et al., 2023b) packages. The R script used for plotting these results are available in Zenodo by Carrión-Olmedo (2025).

We rooted the phylogeny with representative samples of Lestoros inca and Rhyncholestes raphanurus. The inferred phylogenetic trees were visualized and edited using FigTree v1.4.4 (Rambaut, 2018) and Adobe Illustrator 27.9.4.

Species concept and delimitation

We adhere to the De Queiroz’s (2007) concept of species as independently evolving metapopulation lineages. We replicated the methodology proposed by Ruelas et al. (2024) to delimit species with the Assemblage of Species by Automatic Partitioning (Puillandre, Brouillet & Achaz, 2021), and Bayesian implementation of the Poisson Tree Processes (Zhang et al., 2013). The ASAP method was implemented in the web server (https://bioinfo.mnhn.fr/abi/public/asap/) using the default settings and simple distance. Similarly, bPTP was implemented in the web server version (https://species.h-its.org/) with the Bayesian PTP maximum likelihood (bPTP-ML) strategy.

Divergence times

The chronological scheme of South American Land Mammals Ages (SALMA) follows Croft et al. (2009), as adopted by Abello, Martin & Cardoso (2021). We performed divergence time estimation using Bayesian Analysis Sampling Trees BEAST2 (Bouckaert et al., 2019). Initially, sequences alignments were prepared and configured using BEAUti, where we specified unliked site models and unlinked clock models for codon positions as follows: cytb {1+2}, cytb {3}, RAG1 {1+2}, and RAG1 {3}. Site models were evaluated using bModel Test (Bouckaert & Drummond, 2017). An optimised relaxed clock model was used to account for rate variation among lineages (Zhang & Drummond, 2020; Douglas, Zhang & Bouckaert, 2021).

The calibration points were defined based on fossil evidence of extinct members of the Caenolestidae family, sensu Abello, Martin & Cardoso (2021, Fig 4A). Specifically, we used Stilotherium dissimile, Stilotherium parvum, Gaimanlestes pascuali, Pliolestes venetus, Pliolestes triptamicus, and Caenolestoides miocaenicus (Abello, Martin & Cardoso, 2021)—species representing the most recent common ancestors MRCA of extant caenolestids. These fossils span from the Early Miocene, Colhuehuapian (21–17.5 million years ago) to the Late Miocene, Huayquerian (9–6.3 million years ago), providing temporal constraints for the nodes. The Fossilized Birth-Death (FBD) model was implemented to incorporate both extant and extinct taxa in the tree calibration framework (Gavryushkina et al., 2014).

Two independent Markov Chain Monte Carlo (MCMC) analyses were run for 100 million generations each, sampling every 10,000 generations. The resulting log and tree files from each run were combined using LogCombiner v2.6.3 (part of the BEAST2 package), with a burn-in of 10% applied to each file. Convergence and effective sample sizes (ESS) were assessed using Tracer v1.7.2 (Rambaut et al., 2018), ensuring that all parameters had ESS values above 200. The resulting maximum clade credibility (MCC) tree was generated with TreeAnnotator v1.10.4 (Rambaut & Drummond, 2018) and visualized using FigTree v1.4.4 (Rambaut, 2018).

Results

Nucleotide phylogeny

The cytochrome b matrix was 1,143 bp long for 50 individuals (43 sequences generated by us) and the recombination activating gene 1 was 2,838 bp long for 31 individuals (30 sequences generated by us). GenBank accession numbers are listed in Table 1. Partitions and best evolutionary schemes are detailed as Supplemental Material 1. Gene-independent and amino acid phylogenies are available as Supplemental Material 2.

The monophyly in the genus Caenolestes is well-supported with branch-support values of 99.9/100 (Fig. 1). Our mtDNA+nuDNA concatenated matrix analysis recovered two early divergent clades. Clade A contains Caenolestes convelatus + (C. condorensis + (C. sangay + (C. sp. 1 “Loja” + (C. sp. 2 “Cajamarca” + C. caniventer)))) and shows branch-support values of 72.5/79. The Clade B contains taxa identified as “C. fuliginosus” and shows a branch-support values of 99.8/100, however, our results suggest the presence of three to four species within “C. fuliginosus” name (Fig. 1).

Figure 1 Maximum likelihood tree based on concatenated matrix of cytb+RAG1 DNA sequences of the family Caenolestidae.

Branch values indicate SH-aLRT support (%)/ultrafast bootstrap support (%). Terminals in red correspond to sequences from previous studies; terminals in black represent newly generated sequences from this study. Each terminal is labeled with the corresponding voucher number and locality. Column titles refer to currently described species and species delimitation methods applied: ASAP (Assemble Species by Automatic Partitioning) and bPTP-ML (Bayesian implementation of the Poisson Tree Processes using maximum likelihood input). The scale bar represents substitutions per site.

The widely distributed Caenolestes fuliginous showed a previously unknown phylogenetic structure and complexity. Firstly, C. fuliginosus is divided into two clades. The first clade presents low branch support values of 0/43. Despite the low support of the deep branch, we recovered two sister subclades with 100/100 branch support each. One subclade, here mentioned as C. sp. 3 “Baños”, from Tungurahua and the second subclade, here mentioned as C. sp. 4 “Imbabura”, from Napo, Imbabura, and Carchi. It is noteworthy that both clades are mainly distributed in the eastern slopes of the Andes. The second clade shows higher branch support values of 80.1/65. Within this clade, we recovered three subclades. Caenolestes sp. 5 “Golondrinas” with 99.9/100 branch support is the first to diverge. Then, C. sp. 6 “Toisan” is recovered as the sister clade of C. fuliginosus sensu stricto. These taxa are mainly distributed in the western slopes of the Andes.

Additional evidence of the molecular complexity within Caenolestes fuliginosus was obtained from the p-distances (Fig. 2). Within the clade named under Caenolestes fuliginous, taxa have distances of up to 11.23%, (see Supplemental Material 3) suggesting that C. fuliginosus is a species complex.

Figure 2 Pairwise genetic distance matrix and species delimitation among Caenolestes and related taxa.

The heatmap shows pairwise uncorrected p-distances among mitochondrial sequences, with darker colors representing lower divergence and lighter colors indicating higher divergence. The histogram below represents the frequency distribution of pairwise distances across all comparisons. Terminal labels correspond to voucher numbers and localities; sequences in red are from previous studies, while those in black were newly generated in this study. The rightmost color-coded column groups sequences according to currently recognized species or candidate species based on species delimitation methods. Taxonomic assignments reflect results from ASAP (Assemble Species by Automatic Partitioning) and bPTP-ML (Bayesian Poisson Tree Processes using maximum likelihood input).

Species delimitation

Analysis of species delimitation (Fig. 1) recovered 12 and 13 putative species for the Assemblage of Species by Automatic Partitioning (Puillandre, Brouillet & Achaz, 2021) and the Bayesian implementation of the Poisson Tree Processes (Zhang et al., 2013) respectively. This analysis suggests a cryptic diversity within Caenolestes fuliginosus and C. caniventer, as four and two new lineages were recovered respectively.

Aminoacids phylogeny

The amino acids concatenated matrix (Fig. 3) shows a different and more conservative scenario, where less lineages are recovered as independent. The amino acids from cytb and RAG1 contains 42 and 18 informative sites respectively. Partitions and best evolutionary schemes are detailed as Supplementary Material 1. This inference recovered C. convelatus as the first lineage to diverge with 98.1/96 branch support. Then, the Clade A (C. condorensis, C. sangay, and C. caniventer) is recovered with 95.6/97 branch support. The clade B ((C. fuliginosus) + (C. sp. 5 “Golondrinas” + (C. sp. 3 “Baños” + C. sp. 4 “Imbabura”)) is recovered with 95.9/95 branch support.

Figure 3 Maximum Likelihood tree based on Amino acid tree depicting all extant species of Caenolestidae.

Branch values indicate SH-aLRT support (%)/ultrafast bootstrap support (%). Terminals in red correspond to sequences from previous studies; terminals in black represent newly generated sequences from this study. The scale bar represents substitutions per amino acids.

Divergence times

The divergence estimates suggest that the ancestor of Caenolestes existed during the middle Miocene approximately 11 million years ago (Fig. 4A). The diversification within Caenolestes occurred mainly during the middle to late Miocene and into the Pliocene, aligning with uplift phases of the Andes (Fig. 4C).

In Fig. 4B reveals that recently diverged Caenolestes species occupy different elevational bands. Closely related species are generally not sympatric in altitude. For example, Caenolestes fuliginosus is restricted to montane forests and Paramo, while its sister taxa (C. sp. 5 Carchi and C. sp. 6 Toisan) are found exclusively in lower elevations of the cloud forests. Similarly, C. sp. 3 Tungurahua occupies mid-elevation montane forests and paramos, whereas its closest relative, C. sp. 4 Imbabura, is found at lower elevations of cloud forests. This suggests allopatric speciation across an altitudinal gradient that coincide with estimated times of Andean uplift (Fig. 4C). It is remarkable to note that the clade identified as Caenolestes fuliginosus is at least as old as the clade that includes C. condorensis + (C. sangay + (C. sp 1+ C. caniventer)). The materials supporting divergence time estimation of Caenolestes are presented in detail in Supplemental Material 4, including estimated mean node ages and 95% highest posterior density (HPD).

New species delimitations suggest strong endemism within Caenolestes, this pattern coincide with already identified pockets of endemism for other taxa as the upper Pastaza river basin (e.g., Jost, 2004; Reyes-Puig & Yánez-Muñoz, 2012; Freile, Pardo-González & Ordóñez Delgado, 2022; Reyes-Puig et al., 2022b; Reyes-Puig et al., 2022a; Reyes-Puig et al., 2023; Reyes-Puig et al., 2024; Ortega, Brito & Ron, 2022; Clark et al., 2023; Ríos-Alvear et al., 2024), Mira River Basin-Cerro Golondrinas (e.g., Yánez-Muñoz et al., 2018; Yánez-Muñoz et al., 2021b; Yánez-Muñoz et al., 2021a; Yánez-Muñoz et al., 2025; Brito et al., 2020; Reyes-Puig et al., 2020), Cotacachi-Cayapas (C. sp. 6 Toisan) (e.g., Brito et al., 2020; Brito et al., 2024), and Yacuri & Podocarpus National Parks (C. sp. 1 Loja) (e.g., Carvajal-Endara et al., 2019) (Fig. 5).

Figure 4 Chronogram for the family Caenolestidae obtained under a relaxed clock model based on cytb and RAG1.

(A) Divergence times estimated for caenolestids. Age estimates, including maximum and minimum intervals for all nodes, are provided (B) Simplified phylogenetic hypothesis and elevation distribution of recovered lineages. Above, mean elevation of the Andes along Miocene, Pliocene and Pleistocene. Reconstructed and adapted from Pérez-Escobar et al. (2017).

Figure 5 Geographic distribution of Caenolestes species and candidate lineages in the Ecuadorian Andes.

Symbols represent sampled localities, color-coded according to species or candidate species inferred from species delimitation analyses (see inset tree and legend). Black symbols correspond to described species, while colored symbols indicate newly sampled lineages identified in this study. Elevation is represented in shaded relief, and major rivers and volcanoes are labeled. The inset shows a simplified phylogenetic tree with corresponding color codes used for each clade or lineage. The small map in the upper left corner shows the location of Ecuador within South America, with the study area highlighted in red. Map includes administrative boundaries and major hydrographic features for spatial context.

Discussion

Phylogenetic remarks

Caenolestes convelatus

We expand the molecular sampling for this species. The only available cytb sequence for this species was released by Ojala-Barbour et al. (2013). KF418782 accession is an incomplete 759 bp sequence with a 253 bp gap in the middle of the sequence. We made available complete 1,143 bp sequences of cytb and 2,838 bp sequences of RAG1 from two specimens of the same locality. Ojala-Barbour et al. (2013) placed to Caenolestes convelatus as the sister species of C. fuliginosus in their amino acids tree Fig 7A, while in the amino acids + morphology tree Fig 7B placed as the first species to diverge in the genus. Our nucleotide phylogeny (Fig. 1) placed it in the same clade as C. condorensis, C. sangay, and C. caniventer, while our amino acid phylogeny (Fig. 3) is concordant with Fig 7B of Ojala-Barbour et al. (2013).

Caenolestes condorensis

Our results (Figs. 1 and 3) places to Caenolestes condorensis as a sister species of (C. sangay + C. caniventer) for the first time in a molecular phylogeny. This shows concordance to the reported relationships in Fig 7B by Ojala-Barbour et al. (2013), who used only morphological characters.

Caenolestes caniventer

In the nucleotide phylogeny (Fig. 1) the specimen AMNH 268103 from Peru, reported as C. caniventer by Ojala-Barbour et al. (2013) is recovered as a possible undescribed species. This is supported by its long branch and the cytb p-distances in a range between 5% to 7.5% for C. caniventer from Peru and C. caniventer from its type locality (Fig. 2). However, our amino acids analysis (Fig. 3) did not recover it as a possible undescribed species as the nucleotide inference. We suggest this specimen status requires further study as its branch support is low 79.5/76. It is noteworthy that we were not able to sequence the RAG1 fragment for this specimen, so the phylogenetic position and the branch support values might change with new information.

Cryptic diversity in Caenolestes fuliginosus

The previous few decades of molecular studies on small vertebrates, and other taxa, have speed up the detection of cryptic species. Nominal species have often been shown to encompass more undescribed species within their name (Ceballos & Ehrlich, 2006; Hending, 2025) and those discovered species inhabit more restricted areas than previously thought (Bickford et al., 2007). Under these premises and based on our understanding of the Caenolestes genus, we suspected that a thorough revision was needed, especially in widely distributed species like Caenolestes fuliginosus. The published geographic distribution for C. fuliginosus (Martin, 2016; González et al., 2024) suggested to us that this was an undescribed species complex. However, it was not until our molecular analyses that an estimation of its richness was possible, and we were able to uncover three or four possible undescribed species within C. fuliginosus.

The level of genetic divergence observed among samples identified as Caenolestes fuliginosus exceeds commonly used thresholds for species delimitation in small mammals, such as mitochondrial divergences > 5% for cytochrome b (Baker & Bradley, 2006). In some cases, divergence values within “C. fuliginosus” approach or exceed 7–10% (Fig. 2), suggesting that these lineages represent independently evolving units that meet the criteria of De Queiroz (2007) unified species concept and therefore warrant taxonomic recognition. This magnitude of divergence is consistent with patterns reported in other studies describing cryptic rodent or marsupial species in the Neotropics (e.g., Voss, Giarla & Jansa, 2021; Brito et al., 2024; Ruelas et al., 2024).

From our results, nucleotide and amino acids phylogenies, C. fuliginosus appears to be a species complex, comprising at least three undescribed species. Both analyses recovered C. sp. 3 “Baños”, C. sp. 4 “Imbabura”, and C. sp. 5 “Golondrinas”. The only discordance between the nucleotide and amino acid trees lies in the placement of C. sp. 5 “Golondrinas”. In the nucleotide-based analysis, C. sp. 5 “Golondrinas” is the sister species to C. fuliginosus, whereas in the amino acid–based analysis, it is recovered as the sister species to the clade formed by C. sp. 3 “Baños” and C. sp. 4 “Imbabura”. Similar patterns are observed in the efforts made by González et al. (2024) for Caenolestes fuliginosus in Colombia. Likewise, diversity in C. caniventer is still far from being fully understood. Undoubtedly, it is needed additional molecular sampling efforts in Colombia and Peru to resolve the richness of these clades.

Ecological types

It’s been a century since Anthony (1924) recognized two morphologically and ecologically distinct groups within Caenolestes. The small, gracile and delicate shrew-opossum of C. fuliginosus and the large shrew-opossum group formed by C. convelatus, and C caniventer (later joined C. condorensis, and C. sangay). Our nucleotide analysis (Fig. 1) recovered these two groups as two clades. At first glance, we could validate this morphological and ecological groups as the nucleotide inference (Fig. 1) separates them, however, clade A shows a low branch support of 72.5/79, meanwhile, amino acids inference (Fig. 3) excluded Caenolestes convelatus from the clade A and branch supports are higher. C. convelatus shows 98.1/96 branch supports and Clade A shows 95.6/97 branch supports.

Additionally, elevation distributions noted by Anthony (1924) that placed Caenolestes fuliginosus from paramo to high elevation cloud forests and the other species restricted to middle elevation cloud forest and subtropical habitats are not concordant with our elevation distribution data (Fig. 4B). For example, C. caniventer is distributed from 1,700 m to 3,900 m showing the widest elevational distribution. In the case of the species within the C. fuliginosus name (e.g., C. sp. 4 “Imbabura” and C. sp. 5 “Carchi”) are distributed in middle elevation cloudforests.

Andean uplift

Our estimations of divergence times are concordant to the Northern Andes Uplift (10–5 mya), a major orographic event that has shaped and impacted the diversification on numerous Neotropical lineages (Hoorn et al., 2010; Pérez-Escobar et al., 2017). This geological process appears to have driven the cladogenesis within the genus, promoting allopatric divergence across elevational gradients. We found non-overlapping altitudinal distribution among sister species of Caenolestes fuliginosus clade B (Fig. 4B) suggesting that vertical niche partitioning likely played a key role in reducing gene flow and facilitating speciation (Rahbek et al., 2019). However, the Northern Andes uplift (10–5 mya) does not fully explain the cladogenesis within Clade A. While C. convelatus is distributed in the northwestern Andes, the other species in Clade A (C. condorensis, C. sangay, and C. caniventer) are restricted to the southeastern Andes. We hypothesize that this disjunct distribution arose due to ecological competition or niche occupation by the ancestor of Clade B in the central Andes of Ecuador, which may have acted as a biotic barrier—isolating C. convelatus to the northwest while confining the remaining Clade A species to the southeast.

This hypothesis is supported by divergence time estimates, which suggest that the timing of cladogenesis in Clade A coincides with the proposed period of lineage separation and potential ecological interference by the Clade B ancestor. This implies that biotic interactions, alongside geological processes, contributed to the spatial diversification of these lineages. Additionally, we propose an alternative hypothesis for the cladogenesis within Caenolestes. Instead of the ancestor of Clade B acting as an ecological barrier that excluded C. convelatus to the northwest and the remaining Clade A species to the southeast, we suggest that the Andean uplift itself functioned as an abiotic filter. The emergence of new high-elevation environments may have prevented the expansion of the large-bodied, lowland-adapted ancestors of Clade A into these newly formed highlands. In contrast, the ancestor of Clade B may have possessed ecological traits that enabled it to colonize and diversify within these higher elevation habitats. These scenarios highlights how both geological processes and ecological filtering may have shaped the spatial diversification and current distribution patterns within the genus.

Limitations and discordance between unilocus vs. multilocus, and nucleotide vs. amino acids approaches

Gene trees are criticized due to potential inaccuracies and their frequent discordance between species trees (Li, 2006; Knowles & Carstens, 2007). Therefore, a multilocus approach is recommended to more accurately infer phylogenetic relationships (Knowles & Carstens, 2007). This issue is exemplified in our analysis (see Supplemental Material 2) where cytb and RAG1 gene trees yield different topologies. The phylogenetic position of Caenolestes convelatus is far from being resolved and raises further questions. The Cytb tree places C. convelatus as the earliest diverging lineage, whereas the RAG1 tree recovers it as sister of C. condorensis, C. sangay, and C. caniventer.

This uncertainty increases when nucleotide-based trees are compared to those based on amino acids. Amino acids trees reduce the impact of synonymous substitutions and are commonly used for deeper evolutionary timescales. However, evidence suggests that the nucleotide sequence of cytb has high phylogenetic informativeness for resolving questions of recent vertebrate evolution, particularly for divergences more recent than 60 million years ago (Townsend, López-Giráldez & Friedman, 2008) and the amino acid sequence of cytochrome b did not feature the outstanding peak of informativeness for recent times that characterized its nucleotide sequence (see Fig. 2A of Townsend, López-Giráldez & Friedman, 2008). Further evidence supporting the reduction of variance is detailed in Supplemental Material 1. where the third position of the cytb contains 286 informative sites out of 381 total, and third position of the RAG1 contains 50 informative sites out of 946 total sites. In contrast, the amino acids alignment cytb +RAG1 retained only 42 and 18 informative sites respectively, reflecting a more conserved signal.

Finally, a multilocus phylogeny approach could serve as an exploratory tool for rapidly assessing diversity. Future research should incorporate more comprehensive datasets e.g., complete mitogenomes (Nilsson et al., 2003), SNPs, or Whole Genome studies that could further support or refute our initial hypotheses.

Implications for the conservation of caenolestids

The discovery of additional lineages in what was previously thought as a single species (i.e., Caenolestes fuliginosus) requires conservation reassessments. These new lineages are considered pockets of endemism and need special consideration in conservation planning (Bickford et al., 2007). New lineages will have a subset of the overall geographic distribution assigned to what was believed to be a single species (Fig. 4). Thus, Caenolestes fuliginosus, placed in the “Least Concern” category of the IUCN (Martin, 2016) and Ecuadorian Mammals Red List (Tirira, 2021) must be reevaluated.

Our results suggest at least three lineages requiring potential species recognition with restricted distributions. Examples of cryptic species and species complex with previously unknown lineages with restricted distributions, that sum up to biodiversity but require conservation reassessments are numerous among small vertebrates in South America (e.g., mammals: Jarrín & Kunz, 2011; Rocha et al., 2018; Brito et al., 2022a; Brito et al., 2022b; reptiles: Yánez-Muñoz et al., 2018; and amphibians: Páez & Ron, 2019; Funk, Caminer & Ron, 2012).

Detecting cryptic species and determining their distribution ranges has proven important for conservation policies, as such discoveries can lead to the creation of private reserves that protect the only known localities of new species and guide government efforts to expand the National System of Protected Areas (Sistema Nacional de Áreas Protegidas, SNAP –MAATE). For example, the Cerro Candelaria Reserve, a private protected area that harbors multiple endemic species, e.g., Pristimantis loujosti (Yánez-Muñoz, Cisneros-Heredia & Reyes, 2010), P. puruscafeum (Reyes-Puig et al., 2014), and P. normaewingae (Reyes-Puig et al., 2024), has been recognized as part of the SNAP to ensure long-term protection of these ecosystems under government oversight.

Additional cases of cryptic species with are exemplified in the Dracula Reserve, where species such as Anolis dracula (Yánez-Muñoz et al., 2018), Hyloscirtus conscientia (Yánez-Muñoz et al., 2021a), Echinosaura fischerorum (Yánez-Muñoz et al., 2021b), Pattonimus ecominga (Brito et al., 2020), and Chilomys carapazi (Brito et al., 2022b), once considered part of more widely distributed taxa, are now recognized as endemics restricted to the reserve. The discovery and formal description of these species has played a key role in strengthening local conservation actions, supporting the legal protection of the area, and justifying its inclusion within national and international conservation frameworks.

Pending definitions in the taxonomy of the Caenolestes

We have generated a comprehensive phylogeny and identified novel lineages for Ecuadorian caenolestids, but richness estimation and accurate geographic distribution are still incomplete if Colombian and Peruvian samples are not included. We suggest that some lineages might have a transboundary distribution. (e.g., Caenolestes sp. 5 “Golondrinas” with Colombia).

Conclusions

This study adds to our understanding of the evolutionary history of Ecuadorian caenolestids, reveals novel insights on Caenolestes fuliginosus diversity, and suggests the presence of at least three species-level lineages within C. fuliginosus. We refrain from describing these identified species-level lineages until additional morphological evidence becomes available. Quantitative evidence in morphology and osteology are needed to determine diagnostic characters and evaluate synapomorphies for the group. Altogether, this is by far the most taxon-dense, molecularly rich, and geographically well-represented caenolestids dataset ever analyzed.

Supplemental Information

Supplemental Information 1 Bests schemes and partitions

(A) Nucleotide concatenated (Cytb + RAG1). (B) Amino acids concatenated (Cytb + RAG1). (C) Nucleotide Cytochrome b (D) Amino acid Cytochrome b (E) Nucleotide RAG1 (F) Amino acid RAG1

Supplemental Information 2 Individual phylogenetic trees

(A) Nucleotide concatenated (Cytb + RAG1). (B) Amino acids concatenated (Cytb + RAG1). (C) Nucleotide Cytochrome b (D) Amino acid Cytochrome b (E) Nucleotide RAG1 (F) Amino acid RAG1

Supplemental Information 3 Raw table of uncorrected p distances

Uncorrected p-distances (%) based on mitochondrial cytochrome b sequences among samples of Caenolestes. Values below the diagonal represent pairwise genetic distances. The matrix was generated using the software MEGA.

Supplemental Information 4 Materials supporting divergence time estimation of Caenolestes

(A) Table with tip dates and geographic occurrence data for all samples included in the analysis. (B) List of fossil calibration priors used in MCMCTree, including minimum and maximum node age constraints with corresponding references. (C) Dated phylogeny showing 95% highest posterior density (HPD) intervals for each node. (D) Dated phylogeny with estimated mean node ages indicated.

Our gratitude to Rubí García, Jenny Curay, Mishell Noboa, Ana Pilatasig, Ruth Bravo, Rocío Vargas, Jhandry Guaya, Yuleidy Castro, César Garzón, Glenda Pozo, and Hernando Roman for their assistance in fieldwork. We are thankful to Pamela Loján and Daniela Reyes for their invaluable work in the Laboratory of Nucleic Acids at the Instituto Nacional de Biodiversidad (INABIO). Melanie Polo, Laura Simba, Fulton Barros, Dayana Vazquez, Annahy Ayala, Joss Salinas, and Andrés Oña helped during laboratory work. Thanks to Diego Inclán, Francisco Prieto, María Elisa Lara, and Pablo Jarrín-V for their institutional support. Thanks to María José Navarrete for her fruitful conversations and invaluable suggestions included in this research. We are grateful to Lou Jost and Javier Robayo for their tireless efforts to protect the cloudforests where most of the Caenolestes specimens used in this study were found. We are thankful to the Universidad del Azuay. We thank Santiago Burneo, Alejandra Camacho, and Ana Pilatasig from the Museo de Zoología QCAZ, PUCE for facilitating tissues of Caenolestes. We thank Sofía Carvajal from Tandayapa Cloud Forest Station, Universidad San Francisco de Quito. We are deeply indebted to the above-mentioned people and institutions. AI tool ChatGPT was used to detect grammatical errors in the manuscript. Finally, we thank Maria Nilsson, and two anonymous reviewers who have improved the quality of this work.

Additional Information and Declarations

Competing Interests

Author Contributions

Animal Ethics

Field Study Permissions

Data Availability

The authors declare there are no competing interests.

Julio C. Carrión-Olmedo conceived and designed the experiments, performed the experiments, analyzed the data, prepared figures and/or tables, authored or reviewed drafts of the article, and approved the final draft.

Jorge Brito conceived and designed the experiments, performed the experiments, analyzed the data, prepared figures and/or tables, authored or reviewed drafts of the article, and approved the final draft.

The following information was supplied relating to ethical approvals (i.e., approving body and any reference numbers):

We followed the Ministerio del Ambiente, Agua y Transición Ecológica of Ecuador guidelines, No.: 005-2014-I-B-DPMS/MAE, 007-IC-DPACH-MAE-2016, 005-IC-FLOFAU-DPAEO-MAE, 003-2019-IC-FLO-FAU-DPAC/MAE, MAE-DNB-CM-2019-0126, MAAE-ARSFC-2020-0642, MAAE-ARSFC-2021-1644MAATE-ARSFC-2023-0145. Additionally, we followed the handling guidelines of the American Society of Mammalogists Sikes & Animal Care and Use Committee of the American Society of Mammalogists (2016).

The following information was supplied relating to field study approvals (i.e., approving body and any reference numbers):

The Ministerio del Ambiente, Agua y Transición Ecológica of Ecuador approved the study. This study was developed under the research permit No. 005-2014-I-B-DPMS/MAE, 007-IC-DPACH-MAE-2016, 005-IC-FLOFAU-DPAEO-MAE, 003-2019-IC-FLO-FAU-DPAC/MAE, MAE-DNB-CM-2019-0126, MAAE-ARSFC-2020-0642, MAAE-ARSFC-2021-1644MAATE-ARSFC-2023-0145, MAATE-ARSFC-2024-1064, and the authorization for access to genetic resources No. MAATE-DBI-CM-2023-0334.

The following information was supplied regarding data availability:

The data is available at GenBank: PQ570945–PQ570972 and PV505063–PV505107.

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
