# Peer review of "Cryptic diversity on the genus Caenolestes (Caenolestidae: Paucituberculata) in the Ecuadorian Andes"

_PeerJ, doi:10.7717/peerj.19648_

## Round 0.1 · original submission · Major Revisions

Thank you very much for your manuscript titled “Cryptic diversity on the genus Caenolestes (Caenolestidae: Paucituberculata) in the Ecuadorian Andes” that you sent to PeerJ.

This study presents very valuable and relevant information on diversity of marsupial genus Caenolestes. As you will see below, comments from referee 1 suggest a minor revision while reviewers 2 and 3 suggest a major revision before your paper can be published. Given this, I would like to see a major revision dealing with the comments. Their comments should provide a clear idea for you to review, hopefully improving the clarity and rigor of the presentation of your work. I will be happy to accept your article pending further revisions, detailed by to address the observations focused mainly on possible sequencing errors, as well as to clarify some points in the estimation of divergence times for Caenolestes individuals and the new species. It is also necessary to improve some points of the discussion and some observations to improve the figures are also pointed out.

Please note that we consider these revisions to be important and your revised manuscript will likely need to be revised again.

Reviewer 1 ·

Basic reporting

All my comments are written below.

Experimental design

All my comments are written below.

Validity of the findings

Text is well written. All my comments are written below.

Additional comments

Carrión-Olmedo and Brito provide a comprehensive overview of the genetic relationship of the genus Caenolestes (Thomas, 1895). This manuscript is well executed and written. I only have some small remarks to be addressed before publication.

Major comments:
Line 35: An addition of a short description related to the appearance of the animals would be great (e.g. mouse like, opossum like). It comes later in the introduction but earlier would be better so the reader has a better picture of the animals.

Line 45: “Thomas, 1895” Please be consistent with citations/mentioning of the first describers. I would suggest to write them all like (Thomas, 1895). This will improve readability of the text.

Line 121: Please provide a link to the scripts.

Minor Comment:
Line 36: Please rephrase to “and is considered a relictual element in modern faunas”

Reviewer 2 ·

Basic reporting

The manuscript is well-written in clear, with appropriate references providing sufficient context. The structure of the article is ok, and figures and tables are presented clearly. However, some areas require improvement. The Results section presents the findings clearly, supported by appropriate figures and tables, which are easy to interpret. However, the discussion could be improved by addressing all the findings more comprehensively,

Experimental design

The research question is well-defined, addressing an important knowledge gap in the taxonomy and phylogeny of the studied taxa. However, the study would benefit from a clearer explanation of the criteria used for specimen selection and how potential biases in sampling were addressed. Additionally, while the use of cytb data is justified, discussing the limitations of the unilocus approach and suggesting how future genomic or SNP-based studies could complement this work would further enhance the robustness of the design.

Validity of the findings

The findings are robust, supported by statistically sound data, and clearly presented. However, the discussion section requires further development to provide a more comprehensive context for the results. It should incorporate references discussing the biogeographic history of the region, taxa associated with the geographic area, and previous findings that support the presence of cryptic species. This would enhance the scientific depth of the discussion and better connect the findings to the broader literature. Additionally, explicitly addressing the potential impact and novelty of the findings, particularly in terms of conservation and taxonomy, would strengthen the manuscript

Annotated reviews are not available for download in order to protect the identity of reviewers who chose to remain anonymous.

·

Basic reporting

no comment - see below

Experimental design

no comment - see below

Validity of the findings

no comment - see below

Additional comments

The manuscript offers new insights in the diversity of a rarely studied marsupial genus Caenolestes. They sequence and analyze 28 Caenolestes individuals from 20 localities in Ecuador and identify cryptic diversity. Although I am always happy to see phylogenetic studies on marsupials, this study has shortcomings that need to be corrected before publication. Given the severity I recommend major revision.

Main concerns
I strongly suspect that there are sequencing errors in the generated sequences. For this reason, the sequences need to be revised and the data set reanalysed. All the data needs to be closely examined and curated and updated in Genbank in the submitted publicly available sequences.
The reason that some of the sites are potential sequencing errors are:
1) the sequence has a stop codon: this is either a sequencing error or a NUMT. Functional mt genes do not have internal stop codons.
2) the sequence have an amino acid that is different from the rest of the sequences including the outgroup (Lestoros): for instance Caenolestes sp 2, have three different aa in the 5’ of the sequence, these are different to the outgroup as well as another individual of Caenolestes sp 2. Given the high conservation of the sites these must be sequencing errors.
3) the sequence has a one aa deletion that is absent from all other sequences including the outgroup and another conspecic individual: these must be sequencing errors. In general single site aa deletions can occur naturally, but they would not occur between individuals of the same population.

There is a significant amount of natural variation despite the potential sequencing errors in the data set, which indicates cryptic diversity in Caenolestes in Ecuador, however, this may be overstated by the inclusion of sequencing errors. To improve the data and avoid publishing possible mistakes I encourage the authors to closely examine their sequences both on the aa and nt level.

The authors should justify the use of rodent specific primers for the amplification of marsupial cytochrome b (Line 110-111, Smith and Patton 1993). They should add the primer sequences as well as the length of the amplified region to material and methods. The risk with using very degenerate primers is that there could be amplification of NUMTs that influence the downstream analysis. This may or may not be the reason for the sequence errors. They sequenced the PCR products using Nanopore technology which is known to be very error-prone.

The authors have estimated the divergence times for the Caenolestes individuals and new species. They used BEAST, which can be difficult to work with, but is a community standard. They state that they used calibration points from Abello et al 2018 and TimeTree.
Nowhere in the main or supplement is there information about the selected calibration points. The authors need to add a table with the following information for each calibration point: Age (My) and the priors that have been used in BEAST (offset, m, s and median) as well as other settings for the BEAST runs. Without this information it will be impossible to reproduce their study.

Given the age of the included species, I also recommend running the phylogenetic analysis on amino acid sequences as well.

The language is generally fine, but sometimes sentences are unclear. Given that the data need to be reanalysed, their conclusions may slightly change and they need to carefully check the text to remove ambiguous and unclear statements.

The figures are well prepared but need additional information in the figure text to help the reader understand them.
Figure 1: Explain in figure text the numbers below the bars and add ASAP, b-PTP-ML, b-PTP-h. Explanation for scale bar, increase font size of names in the phylogeny.
Figure 2: Larger font size overall. In C, why are the numbers negative?
Figure 4: Explain what EEO stands for, typo in legend (evelation). What does the inset map show? The text needs to be more detailed.

---

## Round 0.2 · accepted · Accept

After reviewing this revised version of your manuscript, I see that the main comments suggested by the reviewers have been included, while the suggestions not considered are justified in detail. Therefore, I am satisfied with the current version and consider it ready for publication.

Reviewer 1 ·

Basic reporting

The authors have addressed my comments and improved the manuscript. I therefore have no further concerns.

Experimental design

See 1

Validity of the findings

See 1

Additional comments

See 1

·

Basic reporting

no comment

Experimental design

no comment

Validity of the findings

no comment

Additional comments

no comment